# Viscoelastic Characterization of a Thermoplastic Elastomer Processed through Material Extrusion

**DOI:** 10.3390/polym14142914

**Published:** 2022-07-18

**Authors:** Bàrbara Adrover-Monserrat, Silvia García-Vilana, David Sánchez-Molina, Jordi Llumà, Ramón Jerez-Mesa, J. Antonio Travieso-Rodriguez

**Affiliations:** Universitat Politècnica de Catalunya, Escola d’Enginyeria de Barcelona Est, Av. d’Edurad Maristany, 10-16, 08019 Barcelona, Spain; barbara.adrover@upc.edu (B.A.-M.); silvia.garcia.vilana@upc.edu (S.G.-V.); david.sanchez-molina@upc.edu (D.S.-M.); jordi.lluma@upc.edu (J.L.); ramon.jerez@upc.edu (R.J.-M.)

**Keywords:** viscoelasticity, mechanical properties, PEBA, constitutive models, prony series

## Abstract

*Objective*. We aim to characterize the viscoelastic behavior of Polyether-Block-Amide (PEBA 90A), provide reference values for the parameters of a constitutive model for the simulation of mechanical behaviors, and paying attention to the influence of the manufacturing conditions. *Methods*. Uniaxial relaxation tests of filaments of PEBA were used to determine the values of the parameters of a Prony series for a Quasi-Linear Visco-Elastic (QLVE) model. Additional, fast cyclic loading tests were used to corroborate the adequacy of the model under different test criteria in a second test situation. *Results*. The QLVE model predicts the results of the relaxation tests very accurately. In addition, the behavior inferred from this model fits very well with the measurements of fast cyclic loading tests. The viscoelastic behavior of PEBA under small strain polymer fits very well to a six-parameter QLVE model.

## 1. Introduction

Additive Manufacturing (AM) technology has emerged as a revolutionary technology in recent history, which consists of obtaining 3D objects by depositing a raw material layer by layer. Currently, there are numerous AM techniques, with Material Extrusion (MEX) being one of the most common. MEX is based on the use of a thermoplastic matrix to ensure the printability of the filament, making it possible to manufacture complex geometries. When manufacturing with MEX technologies, the mechanical properties of the printed samples depend on manufacturing parameters such as printing velocity or temperature.

Numerous studies investigated the different combinations of manufacturing parameters regarding the final mechanical behavior of the samples [1,2]; in particular, combinations of layer height, build orientation, manufacturing velocity, and temperature are the most commonly analyzed. Due to the versatility of MEX for manufacturing different geometries, this technology is used in several applications, such as the manufacture of conductive carbon nanomaterials and metals mixed with a thermoplastic [3], biomedical implants made of thermoplastics with high-temperature stability and high mechanical strength such as PAEK [4], and soft robotics by using flexible filament or thermoplastic elastomers (TPE) [5]. In fact, many research groups are focusing their studies on TPEs due to their properties in flexibility and their good mechanical behavior. Depending on their polymer base, TPEs can be divided into TPE-O (based on olefin), TPC (based on polyester), TPU (based on polyurethane), and PEBA (based on polyamide), among others.

However, despite these TPE’s benefits and their increasing use, further research is required to define its manufacturing conditions with MEX; although manufacturing with this technique is easy, its mechanical behavior may vary concerning common thermoplastics given the TPE’s low stiffness. Furthermore, it is challenging to achieve a complete study of this family of materials because of its complex structure and behavior. For this reason, some recent research studies have investigated thermoplastic elastomers in terms of printability and their possible applications [6], the effect of manufacturing parameters on the mechanical responses [7], and the production of TPEs, as well as the feasibility of using these materials and the viscous response of a TPE after 3D extrusion [8]. Moreover, inter- and intra-layer bonding are studied since the stronger these connections are, the greater the mechanical behavior of the pieces. Assuming that the viscoelasticity of TPEs can affect the manufactured components, the effect of manufacturing parameters that can influence the process of inter- and intra-layer bonding (such as extrusion temperature and manufacturing velocity) has been studied [9,10].

In particular, the dynamic response of TPEs is of great interest because of their frequent use as vibration dampers or in sports equipment, among others. In this field, Polyether-Block-Amide (PEBA) has been used in components and midsoles of high-end shoes (running, football, basketball, trekking, etc.) due to its low specific weight and damping properties, which increases the interest in this material. Because of its wide use, the mechanical and thermomechanical properties of PEBA have been previously reported by some authors [11,12,13,14,15].

However, to fully understand the performance of thermoplastic elastomers, the viscoelastic characterization of its mechanical behavior must be carried out. Further investigation is needed due to their elaborated behavior given by its two-phase structure of hard segments (owing to the thermoplastic properties) and soft segments (given its elastomeric properties), as well as due to its strain-rate-dependent behavior.

Viscoelasticity has been studied for some elastomers, concluding the importance of the viscoelastic contribution [5,8,16,17]. For example, studies of relaxation tests of ethylene–octene copolymer (TPE-O) showed that stress decay achieves the 25% within short-term relaxation [18]. Other authors proposed micromechanical models of TPE-O [19] or models based on FEM simulations for TPU manufactured with laser sintering techniques [20], observing a time-dependent behavior. Moreover, TPU specimens manufactured with the MEX technique showed that their storage modulus depends on the manufacturing temperature [21]. In the study of PEBA, other authors studied the viscoelasticity by using cyclic loading, which leads to the use of the formalism complex-dynamic modulus based on storage modulus $E^{\prime }\left(\omega \right)$ and loss modulus $E^{\prime \prime }\left(\omega \right)$ [11,13,22]. All these studies show the difficulties in representing the viscoelastic behavior of elastomeric materials.

In this piece of work, a viscoelastic characterization of PEBA is developed using an alternative approach based on the Prony series and provides accurate parameters for a Quasi-Linear Visco-Elastic (QLVE) constitutive model for PEBA manufactured by MEX. The same approach has also been used by other authors for the characterization of TPEs [23,24], as well as for other biological materials [25]. The main aim of this paper is to study the viscoelasticity properties, which are also used to predict the behavior of polyether-block-amide shore 90A (PEBA 90A). Moreover, the effect of two manufacturing parameters (temperature and velocity) on the properties mentioned above are investigated to seek a relationship between manufacturing parameters and viscoelastic properties. A comparison of the experimental results and the model predictions is performed in order to validate this work.

## 2. Material, Experimental Procedure and Constitutive Model

### 2.1. Materials and Manufacturing Process

For this study, a polyether-block-amide-based polymer (PEBA 90A) was used. The filament was supplied by Fillamentum^®^ (Hulín, Czech Republic), and the samples were 3D printed. This material is a TPE developed for processing by 3D printing (specifically for material extrusion, also known as Fused Filament Fabrication (FFF)).

For the dynamic tests, the specimens used were PEBA 90A filaments printed with an Ender Pro-3 from Creality^®^ (Shenzhen, China). An open-source software, Ultimaker Cura^®^ from Ultimaker B.V. (Utrecht, The Netherlands), was used to define the processing parameters and to slice the samples. All filaments were printed with a nozzle diameter of 0.40 mm, a width of 0.48 mm, and a layer height of 0.20 mm. The manufacturing temperature and velocity were variable as it was intended to study their possible effect on the microstructure of the samples and the viscoelastic properties, following a methodology similar to some previous studies [26]. Thus, two manufacturing velocity values (1300 mm/min and 2500 mm/min), as well as two manufacturing temperatures (225 °C and 245 °C, being the limits of the recommended range before degradation), were used. The manufacturing parameters were selected according to PEBA’s thermal and physical properties and by considering the range of recommended parameters. Specifically, its printing temperature range is between 225 and 245 °C to ensure that there is no thermal degradation while extruding the filament. The printing velocity studied (1300 and 2500 mm/min) is set to guarantee the accurate printability of the samples. Since the material used is flexible, it is important to verify the manufacturing parameters used as they can lead to extrusion problems when printing as it can generate buckling in the gap between the gears and the hotend. Therefore, four different manufacturing combinations were used to modify these two conditions, and for each condition, three specimens were printed, obtaining a total of 12 specimens.

### 2.2. Experimental Tests

For the dynamic tests, PEBA specimens were tested in the uniaxial tensile loading of two different assays: stress relaxation tests along with fast cyclic loading tests. On the one hand, the relaxation tests are used for fitting the viscoelastic properties and consist of applying a strain to the specimen until a maximum strain level (which was previously defined). Once achieved, this strain level is maintained, where it can be observed that the force decays due to the specimen relaxation. On the other hand, the fast-cycling loading tests allow the corroboration of the model in a different load condition [25,27], and they are based on loading and unloading the specimen at a high strain rate, achieving a maximum previously define strain in each cycle. All these tensile tests were performed using a universal testing machine (UTM) ZWICK^®^ all around 5 kN, and loads were measured with a load cell of 50 N.

The maximum strain for all tests was selected to be 5%. This low-strain level ensures that the deformation takes place in the reversible elastic region. The values of strain and stress on each sample were digitally controlled and saved at a data acquisition frequency of 667 Hz with the aim of increasing the accuracy of the results. The strain rate of the loading and unloading was $\dot{\varepsilon }=0.30$ s${}^{-1}$ (50 mm/s), and the waiting time between the relaxation test and the cyclic test was 600 s to ensure the complete recovery of the specimen. The comparison of both relaxing and cyclic tests is presented in Section 3 where the adequacy of the proposed constitutive model under different mechanical loadings can be observed.

From the force recorded for each time, axial stress is computed as the second Piola–Kirchhoff stress:(1)$$\sigma \left(t\right)=\frac{F_{t}}{\lambda_{t}A_{0}}$$
with $F_{t}$ being the instantaneous force, $A_{0}$ being the initial cross-section of the filament, and $\lambda_{t}=1+\delta_{t}/\ell_{0}$ being the stretch and where $\delta_{t}$ is the displacement applied and $\ell_{0}$ is the initial length. Note that all the non-axial components are zero due to the uniaxial tensile configuration. Using the Green–Lagrangian strain tensor $\varepsilon \left(t\right)=1/2\left(F_{t}^{T}F_{t}-I\right)$ as a measure of strain, the longitudinal strain of the filaments provided by the following:(2)$$\varepsilon \left(t\right):=\varepsilon_{xx}=\frac{1}{2}\left(\lambda_{t}^{2}-1\right)$$
where the other two non-null components are $\varepsilon_{yy}=\varepsilon_{zz}=-\bar{\nu }\left(\lambda_{t}\right)\left(\lambda_{t}^{2}-1\right)$; $\bar{\nu }\left(\lambda_{t}\right)$ represents the Poisson effect (see other studies as [25] for the tensor deduction).

### 2.3. Constitutive Model

For a very low strain rate in the reversible deformation regime, PEBA filaments exhibit a mechanical behavior that can be represented by a hyperelastic constitutive model. However, when the strain is moderately high, the thermoplastic material (PEBA) presents viscoelastic effects that cannot be neglected; therefore, hyperelastic models are insufficient.

In order to represent the effect of strain rate, a viscoelastic extension of a hyperelastic model was considered; in particular, a model of type QLVE [11,28,29] is proposed. In a general QLVE model, the mechanical response is divided into an strain-rate-independent elastic part and a strain-rate-dependent viscous part: $\sigma =\sigma^{\left(e\right)}+\sigma^{\left(v\right)}$. Using this assumption, the axial strain (using the second Piola–Kirchhoff stress tensor) is expressed as follows:(3)$$\sigma \left(t\right)=\sigma^{\left(e\right)}\left(t\right)+\int_{-\infty }^{t}R\left(t-\tau ,\varepsilon \right)\dot{\varepsilon }\left(\tau \right)\;d\tau$$
where the integral provides the viscoelastic contribution to total stress σ(t), and $\sigma^{\left(e\right)}\left(t\right)$ is the purely elastic contribution; moreover, $\dot{\varepsilon }$ represents the strain rate and R(t,ε) is the relaxation function. A separable relaxation function $R\left(t,\varepsilon \right)=G\left(t\right)\cdot \partial \sigma^{\left(e\right)}/\partial \varepsilon$[28,30] is used, where G(t) is the relaxation modulus, which is a decreasing function of time that can be adequately represented by a Prony series [31,32,33]:(4)$$G\left(t\right)=\left\{\begin{array}{cc} \sum_{k=1}^{N}g_{k}e^{-t/\tau_{k}} & t\geq 0\\ 0 & t<0 \end{array}\right.$$
with $g_{k}$ denoting the “weight” of each *k*-term and $\tau_{k}$ denoting its relaxation time. All the above parameters are obtained by conducting experimental relaxation tests. The value of *N* depends on the needs of the setting. In Section 3, it is shown that N=3 is sufficient for representing the data in this study, as it is shown in some previous research studies [25]. In fact, Beykin and Monzón (2005) discussed the approximation of functions by a sum of exponentials [34]; because of their arguments, the specific choice (Equation 4) is completely general for a decreasing function G(t).

Using the above Prony series and the elastic Young’s modulus *E*, Equation (Equation 3) can be explicitly rewritten for low strain as follows.
(5)$$\sigma \left(t\right)=E\varepsilon \left(t\right)+\sum_{k=1}^{3}g_{k}\int_{0}^{t}e^{-\left(t-\tau \right)/\tau_{k}}E\dot{\varepsilon }\left(\tau \right)\;d\tau$$

For viscoelastic problems where cyclic loading occurs, it is useful to calculate complex modulus $\hat{E}\left(\omega \right)$ and the associated real and imaginary parts: storage modulus E′(ω)=ReE^(ω) and loss modulus E″(ω)=ImE^(ω) (with ω being the frequency). The complex modulus is related to the parameters of the Prony series (Equation 4) by the Fourier transform:(6)$$\hat{E}\left(\omega \right)=E\;F\left[G\left(t\right)\right]=E\int_{-\infty }^{+\infty }G\left(\frac{\phi }{\omega }\right)e^{-i\phi }\;d\phi$$
where the phase angle is ϕ=ωt (or t=ϕ/ω). This last expression is equivalent to the following explicit formulas [35].
(7)$$E^{\prime }\left(\omega \right)=E\sum_{k=1}^{N}g_{k}\frac{\omega^{2}\tau_{k}^{2}}{1+\omega^{2}\tau_{k}^{2}},\;E^{\prime \prime }\left(\omega \right)=E\sum_{k=1}^{N}g_{k}\frac{\omega \tau_{k}}{1+\omega^{2}\tau_{k}^{2}}$$

## 3. Results

Regarding the relaxation tests, the parameters obtained from the viscoelastic model fittings for the 12 specimens are shown in Table 1. For each specimen, the model summation has been extended up to k=3, obtaining a remarkable improvement in quality fitting (*R*${}^{2}$ = 0.999) concerning lower *k* values (see Figure 1). Thus, the six parameters ($g_{1},g_{2}$, and $g_{3};\tau_{1},\tau_{2}$, and $\tau_{3}$) of the viscoelastic model were obtained, and the results are shown for each manufacturing temperature and velocity condition (three specimens of each couple of conditions were tested and fitted). The averages for the four conditions are shown in Table 2.

As observed in Table 2, the three characteristic times $\tau_{k}$ represent three different times scales $\tau_{1}=O\left(10^{2}\right)$ s, $\tau_{2}=O\left(10^{1}\right)$ s, and $\tau_{3}=O\left(10^{0}\right)$ s; this is a typical situation in the approximation of the relaxation modulus by means of the Prony series [35,36]. On the other hand, the weights $g_{k}$ of these three scales are similar. No significant difference has been observed in parameters $\tau_{k}$ and $g_{k}$ for the four conditions.

For the fast-loading cycles, it is observed that although the maximum strain is equal for all the cycles, the maximum stress (and force) value decreases for each subsequent cycle due to a conditioning effect that can be seen in Figure 2. This conditioning effect has been previously described for other viscoelastic materials [25,36] and leads to defining the Unconditioned Scale Factor for the *n*th cycle USF_*n*_ as follows.
(8)$${\mathrm{USF}}_{n}=\frac{F_{1}}{F_{n}}$$

Since, in the process of cyclic loading, the value of each peak force is less than the value in the previous cycle, the values of USF${}_{n}$ form a strictly increasing sequence in *n*. As shown in Appendix A, the viscoelastic model of Equation (Equation 3) leads to the expectation of a relationship of the following type:(9)$${\mathrm{USF}}_{n}\approx \frac{1+\alpha_{0}}{1+\sum_{k=1}^{N}\alpha_{k}e^{-\beta_{k}n}}$$
which has the same form as Equation (Equation 18) in the Appendix. Indeed, Equations (Equation 14), (Equation 15) and (Equation 19) together allow us to calculate how the above parameters depend on the values of the previously fitted characteristic times $\tau_{k}$ and weights, computed from the relaxation tests.

Figure 3 shows a comparison of the average of the predicted values and the fitted values directly from the cyclic tests. Predicted values for the parameters are as follows: $\alpha_{0}=0.089\pm 0.016$, $\alpha_{1}=9.2\cdot 10^{-5}\pm 4.9\cdot 10^{-6}$, $\alpha_{2}=1.2\cdot 10^{-3}\pm 1.8\cdot 10^{-4}$, $\alpha_{3}=0.030\pm 0.003$, $\beta_{1}=0.0056\pm 0.0001$, $\beta_{2}=0.093\pm 0.010$, and $\beta_{3}=1.368\pm 0.111$. Because $\alpha_{1}$ and $\alpha_{2}$ are too small and are not very relevant for the comparison, they are not shown in Figure 3.

Finally, the above values of the parameters from Table 2 can be used to find complex modulus $\hat{E}\left(\omega \right)$ and both the associated storage modulus E(ω) and loss modulus $E^{\prime \prime }\left(\omega \right)$ according to Equation (Equation 7). These moduli are represented in Figure 4.

## 4. Discussion

The fittings obtained for the relaxation tests depend on the number of terms used in the Prony series (Equation 4): It was observed that, by using k=3, the fitting performs successfully ($R^{2}>0.999$) for the experimental data. This fact shows that within the considered time range and strain level, a six-parameter QLVE model adequately represents the viscoelastic behavior for viscoelastic stress relaxation. The fast-loading cycle tests represent a very different loading situation, which allowed testing whether the six-parameter QLVE model still made adequate predictions. Experimentally, it was observed that, in these types of tests, the experimental error was considerably larger, which is reflected in a larger scatter of the measurements surrounding the expected value (see Figure 2). That is, a moderate error is present in the measurement of the peak force for each cycle. This can be due to the high speed of the cycles or to the existence of some vibrations of UTM and other uncontrolled factors such as inertial effects [11] that produce some experimental noise. However, even so, the predictions of the model for the expected values of the peak forces are reasonably good ($R^{2}>;0.93$). The value of the predicted parameters for fast-cyclic loading and the fitted values are also adequate, as shown in Figure 3. Nevertheless, the larger experimental errors observed in the fast-cyclic loading tests are reflected in larger standard deviations for the fitted values.

As for the comparison with the QLVE model parameters, most studies on the viscoelastic properties of PEBA use other measurement techniques, with very different frequency ranges; thus, the comparison that can be made is partial. The Young’s modulus of PEBA is of the order of 2 GPa, which leads to the prediction for ω=10 s${}^{-1}$ for the storage modulus to be $E^{\prime }\approx 0.5$ GPa, which is compatible with the results of [37]. However, the value for the loss modulus $E^{\prime \prime }$ differs markedly in both works. This could be because our results cannot be extrapolated to very high frequencies. Moreover, the comparison is difficult since the other work does not provide an experimental error estimate for the viscoelastic moduli [37]. However, the value of the phase angle δ (see Figure 5) in both papers appears to be of the same order of magnitude, 3–14°.

Some limitations of the present work are that measurements should be presented for high frequencies, since the relaxation tests only capture the effect for oscillatory behaviors in the order of 5–10 Hz, due to the time resolution used. Another important factor is that it is well known that, in polymers and composites, the modulus decreases with an increase in temperature [38,39]. Still, this effect has not been investigated in this work; thus, the formulation of a complete model should show how the parameters $\tau_{k}$ and $g_{k}$ vary with temperature. Therefore, further work is needed to clarify this point. In addition, the present work studies homogeneous PEBA filaments, but it should be kept in mind that in the formation of macroscopic parts by MEX, small microdefects may appear that influence the final mechanical failure of the part such that an ambitious model applicable to the simulation of PEBA parts must somehow consider the random details of the microstructure [40,41].

## 5. Conclusions

A QLVE model in which the relaxation function is modeled by the Prony series provides a suitable model to represent the elastic behavior of PEBA. The best method for finding the parameters is to use relaxation tests, which allow experimental determinations to be made with small measurement errors. The inferred results of this model for relaxation tests adequately predict the behavior in other different situations, corroborating the adequacy of the model for the time and strain scales used. No significant influence of the manufacturing parameters used for the fabrication of the specimens was observed.

## Figures and Tables

**Figure 1 polymers-14-02914-f001:**
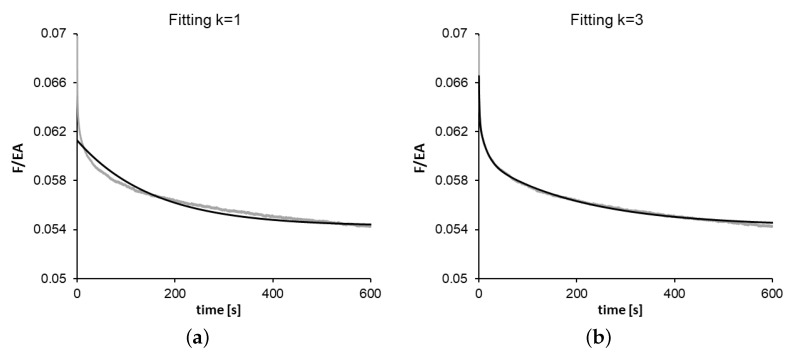
Relaxation data for one of the specimens and QLVE model fitting for (**a**) k = 1 and (**b**) k = 3.

**Figure 2 polymers-14-02914-f002:**
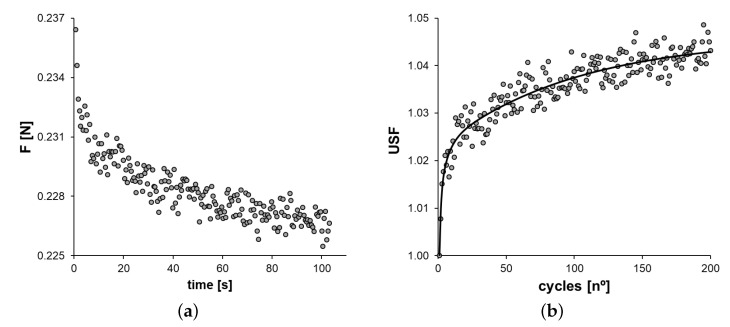
(**a**) Force peak of each cycle during the fast cyclic loading tests at constant displacement and (**b**) unconditioned scale factor (USF) of each cycle (scattering) and USF fitting (solid line) of Equation (Equation 9) for N=3 ($R^{2}=0.936$).

**Figure 3 polymers-14-02914-f003:**
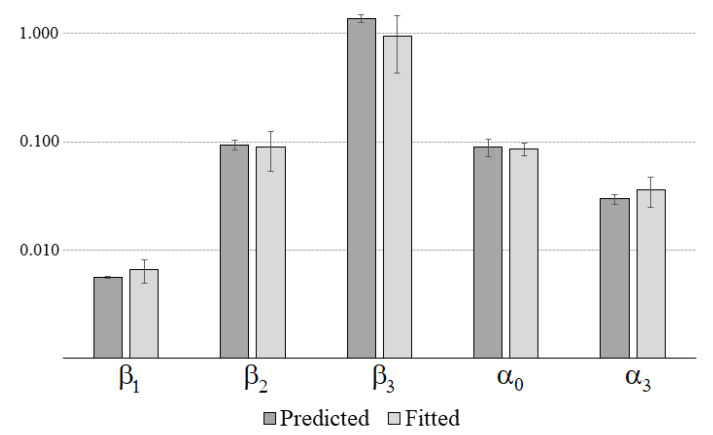
Comparison of the predicted and fitted relevant values for the parameters in the fast-cyclic loading–unloading tests; see Equation (Equation 9).

**Figure 4 polymers-14-02914-f004:**
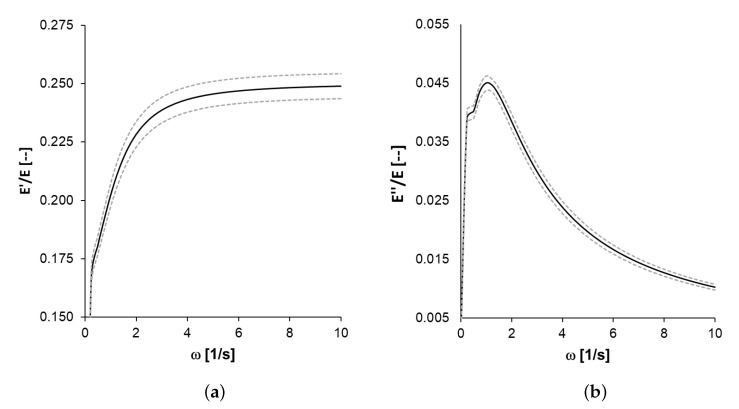
Storage modulus (**a**) $E^{\prime }\left(\omega \right)$ and loss modulus (**b**) $E^{\prime \prime }\left(\omega \right)$ estimated for low frequency form the relaxation parameters.

**Figure 5 polymers-14-02914-f005:**
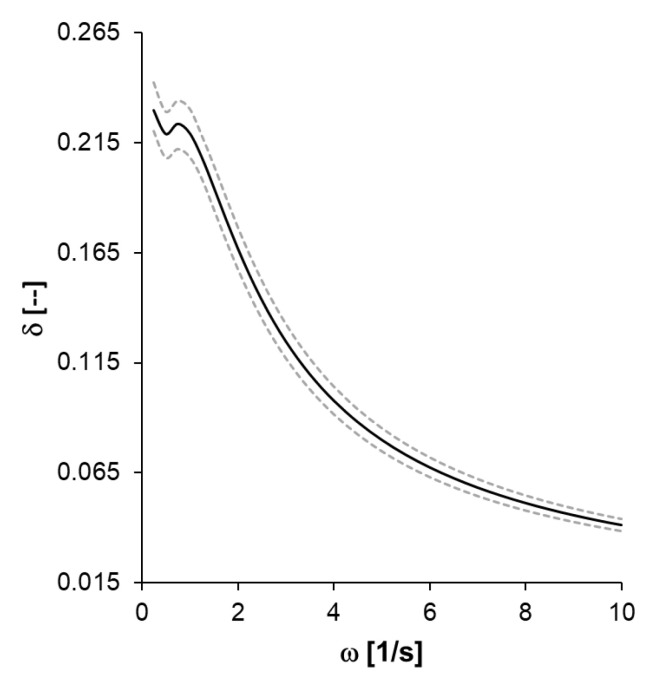
The *phase angle* $\tan\delta =E^{\prime }\left(\omega \right)/E^{\prime \prime }\left(\omega \right)$ predicted from relaxation parameters.

**Table 1 polymers-14-02914-t001:** Parameters obtained from the constitutive model fittings for the specimens, named as temperature (T), speed, and the specimen number.

T-Speed-n^*o*^	$g_{1}$	$g_{2}$	$g_{3}$	$\tau_{1}$	$\tau_{2}$	$\tau_{3}$
**225-1300-1**	0.1014	0.0590	0.0751	203.39	15.27	0.903
**225-1300-2**	0.0987	0.0823	0.0919	194.91	11.60	0.832
**225-1300-3**	0.0996	0.0732	0.0792	201.74	12.05	0.808
**225-2500-1**	0.0997	0.0731	0.0791	198.15	11.81	0.795
**225-2500-2**	0.1031	0.0799	0.0854	198.34	13.25	0.932
**225-2500-3**	0.0961	0.0767	0.0826	200.73	10.97	0.783
**245-1300-1**	0.1025	0.0639	0.0712	191.10	11.70	0.783
**245-1300-2**	0.0957	0.0783	0.0840	196.87	12.51	0.847
**245-1300-3**	0.1049	0.0742	0.0797	193.15	10.58	0.733
**245-2500-1**	0.0948	0.0603	0.0634	198.66	10.20	0.693
**245-2500-2**	0.0975	0.0688	0.0731	202.99	11.08	0.777
**245-2500-3**	0.0950	0.0769	0.0801	194.09	12.13	0.819

**Table 2 polymers-14-02914-t002:** Parameters obtained from the constitutive model fittings for the specimens, named as temperature (T) and speed.

T-Speed	$g_{1}$	$g_{2}$	$g_{3}$	$\tau_{1}$	$\tau_{2}$	$\tau_{3}$
**225–1300**	0.100 ± 0.001	0.072 ± 0.012	0.082 ± 0.009	200.0 ± 4.5	12.97 ± 2.00	0.848 ± 0.050
**225–2500**	0.100 ± 0.003	0.077 ± 0.003	0.082 ± 0.003	199.1 ± 1.4	12.01 ± 1.15	0.837 ± 0.082
**245–1300**	0.101 ± 0.005	0.072 ± 0.007	0.078 ± 0.006	193.7 ± 2.9	11.60 ± 0.97	0.788 ± 0.057
**245–2500**	0.096 ± 0.002	0.069 ± 0.008	0.072 ± 0.008	198.6 ± 4.5	11.14 ± 0.97	0.763 ± 0.064
**average**	0.0991 ± 0.003	0.072 ± 0.008	0.079 ± 0.007	197.8 ± 3.9	11.93 ± 1.35	0.809 ± 0.066

## Data Availability

The data are available in the UPC repository https://upcommons.upc.edu, accessed on 5 November 2021, search under the name of the article.

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
