# Peer review of "Viscoelastic Characterization of a Thermoplastic Elastomer Processed through Material Extrusion"

_polymers, 2022, doi:10.3390/polym14142914_

Round 1

Reviewer 1 Report

This manuscript reports a study of a polyether-block-amide based polymer for its viscoelastic property. Experimental dynamic tests are carried out and a constitutive model of type QLVE is proposed. The viscoelastic behavior of PEBA is found fit well to a six-parameter QLVE model. The paper was overall well written and organized. The results are interesting for the research community, and the topic is in the scope of the journal. I can recommend publication of the paper after addressing the following minor comments/questions.  

Comments:

1.      Most parameters obtained in Table 2 show relative small deviations, but some still has a deviation of more than 10% like tau2. It is not clear how the upper bound and low bound are obtained considering the deviation in Table 2. Could the author discuss more about the effect of deviation from multiple parameters on the uncertainty of the model prediction?

2.      The manufacturing parameters like velocity and temperature affect the mechanical properties of the sample, and the model can be fitted on difference manufacturing cases. 225 and 245 are studied temperatures, and 1300 and 2500 are studied speed. Why these values are of interest is not discussed. How reliable it is to draw a conclusion using only these combinations of T and Speed?

Author Response

(Copy add in pdf, with better quality)

This manuscript reports a study of a polyether-block-amide based polymer for its viscoelastic property. Experimental dynamic tests are carried out and a constitutive model of type QLVE is proposed. The viscoelastic behavior of PEBA is found fit well to a six-parameter QLVE model. The paper was overall well written and organized. The results are interesting for the research community, and the topic is in the scope of the journal. I can recommend publication of the paper after addressing the following minor comments/questions.  

Comments:

  1. Most parameters obtained in Table 2 show relative small deviations, but some still has a deviation of more than 10% like tau2. It is not clear how the upper bound and low bound are obtained considering the deviation in Table 2. Could the author discuss more about the effect of deviation from multiple parameters on the uncertainty of the model prediction?

The values shown in table 2 show “averages ± sample standard deviation” for each parameter. The sample standard deviations for characteristic times  were computed according to the usual formula:

Where  are the different values obtained for characteristic times of each specimen, and are the average for each characteristic time. Four different specimens () were tested for each combination of speed and temperature.

With respect to the calculated standard deviations, it turns out that due to the fitting process, the magnitudes obtained ,  and  are subject to some statistical variation since they are not independent variables, having a small non-zero correlation. This is important, because if the error for  is large as observed for the first condition (225ºC - 1300 mm/min) this can give larger standard deviations in  (although in the 245ºC - 2500 mm/min condition the error in  is not larger than in the other cases, this seems to be just a statistical coincidence). It could be conjectured that at high temperature there is simply greater homogeneity in the manufactured samples, although this has not been corroborated. Anyway, the authors judged that given the value of the deviation with respect to the average, this is a non-significant statistical fluctuation that is to be expected, given the above considerations of small interdependence of the variables.

To demonstrate that, the following figure 1(a) show the experimental curves (black line) and the fitting model in the first specimen 225-1300-1 with the maximum parameter values for this condition (, red line) and the minimum values (, blue line). Figure 1(b) shows the same curve magnified, where no relevant differences are observed (in fact, the blue line was widened with respect to the red one in order to show the slight differences).

Figure 1. (a) Experimental curve of the specimen 225-1300-1 and fitting with the maximum characteristic time values (red) and the minimum values (blue) for this condition. (b) Same curve magnified.

Figure 2 shows the same fitting, using the maximum  values (red line) and the minimum values (blue line) obtained as the mean value of each parameter of 225-1300 condition of Table 2 plus the deviation (maximum) and minus the deviation (minimum) respectively. As shown, the curves are slightly different when the initial region is amplified, even the predicted values less than the 3.3% and only in the first 100 test seconds.

Figure 2. (a) Experimental curve of the specimen 225-1300-1 and fitting with the maximum characteristic time values of Table 2 (red) and the minimum values (blue) for this condition. (b) Same curve magnified.

  1. The manufacturing parameters like velocity and temperature affect the mechanical properties of the sample, and the model can be fitted on difference manufacturing cases. 225 and 245 are studied temperatures, and 1300 and 2500 are studied speed. Why these values are of interest is not discussed. How reliable it is to draw a conclusion using only these combinations of T and Speed?

The manufacturing parameters were selected according to PEBA’s thermal and physical properties and considering the range of recommended parameters. Specifically, its printing temperature range is between 225 and 245 ºC to ensure that there is no thermal degradation while extruding the filament. The printing velocity studied (1300 and 2500 mm/min) is set to guarantee the printability of the samples. Since the material used is flexible, it is important to verify the manufacturing parameters used as they can lead to extruding problems when printing. The initial hypothesis given the prior knowledge of the material suggested that this range would have been sufficient to see if there were significant differences, however, the data showed that the differences were not statistically significant.

The authors would like to thank the reviewer for the review and the pertinent comments. This kind of accurate review is very useful to the entire scientific community. So, thank you very much.

The authors are grateful for the kind comments; some minor modifications have been made to make the text clearer for the reader.

Reviewer 2 Report

The authors proposed a Quasi-Linear Visco-Elastic (QLVE) model to characterize Polyether-Block-Amide.

The article is well-written, and the results support the conclusions.

The methods and the experimental data are robust, allowing for reproduction of them.

The article can be accepted in the present form after improving English. 

Author Response

(Copy add in pdf, with better quality)

Reviewer 2

The authors are grateful for the kind comments; some minor modifications have been made to make the text clearer for the reader (changes not explicitly indicated).

The authors are grateful for the kind comments on their work; some minor modifications have been made to make the text clearer for the reader (changes not explicitly indicated).

Sincerely,

The authors
